# Molecular Spectroscopic Markers of Abnormal Protein Aggregation

**DOI:** 10.3390/molecules25112498

**Published:** 2020-05-27

**Authors:** Natalia Wilkosz, Michał Czaja, Sara Seweryn, Katarzyna Skirlińska-Nosek, Marek Szymonski, Ewelina Lipiec, Kamila Sofińska

**Affiliations:** M. Smoluchowski Institute of Physics, Jagiellonian University, 30-348 Kraków, Poland; natalia.szydlowska@uj.edu.pl (N.W.); michalandrzej.czaja@student.uj.edu.pl (M.C.); saras@poczta.onet.pl (S.S.); katarzyna.skirlinska@gmail.com (K.S.-N.); ufszymon@cyf-kr.edu.pl (M.S.)

**Keywords:** abnormal protein aggregation, secondary structure, amyloids, neurodegenerative diseases, multivariate data analysis, principal component analysis (PCA), hierarchical cluster analysis (HCA), molecular spectroscopy, nanospectroscopy

## Abstract

Abnormal protein aggregation has been intensively studied for over 40 years and broadly discussed in the literature due to its significant role in neurodegenerative diseases etiology. Structural reorganization and conformational changes of the secondary structure upon the aggregation determine aggregation pathways and cytotoxicity of the aggregates, and therefore, numerous analytical techniques are employed for a deep investigation into the secondary structure of abnormal protein aggregates. Molecular spectroscopies, including Raman and infrared ones, are routinely applied in such studies. Recently, the nanoscale spatial resolution of tip-enhanced Raman and infrared nanospectroscopies, as well as the high sensitivity of the surface-enhanced Raman spectroscopy, have brought new insights into our knowledge of abnormal protein aggregation. In this review, we order and summarize all nano- and micro-spectroscopic marker bands related to abnormal aggregation. Each part presents the physical principles of each particular spectroscopic technique listed above and a concise description of all spectral markers detected with these techniques in the spectra of neurodegenerative proteins and their model systems. Finally, a section concerning the application of multivariate data analysis for extraction of the spectral marker bands is included.

## 1. Introduction

Following the increase of global average life expectancy, the world population has experienced an upsurge in the number of individuals suffering from deadly and debilitating neurodegenerative diseases, such as Alzheimer’s, Parkinson’s, and Huntington’s disorders [1,2,3]. Recent estimations imply that over 50 million people worldwide are living with dementia, and the cost of treatment exceed already 1% of the global GDP [4]. Neurodegenerative diseases not only affect individuals and their families, but also pose a crippling burden on the healthcare systems. Despite intense scientific efforts all over the world, dementia is still incurable. The neurodegenerative processes at the heart of those diseases are caused by an abnormal aggregation of highly cytotoxic, structurally pathological proteins called amyloids [1,2,3]. A detailed investigation into the secondary structures of neurodegenerative proteins and model proteins, that undergo fibrillation processes, is crucial for an understanding of the etiology of neurodegenerative diseases, which will allow for the development of successful treatment regimes.

Abnormal aggregation of proteins such as β-amyloid, tau, α-synuclein and polyglutamine-containing proteins is known to be related to the pathology of neurodegenerative diseases including Alzheimer’s disease (AD), Parkinson’s disease (PD) and Huntington’s disease (HD) [5,6,7]. Protein aggregates are toxic and lead to neuronal death in different brain regions depending on the disease [8,9]. The etiology of neurodegenerative diseases has not been explained yet. A lot of scientists working on this issue have suggested various mechanisms of protein misfolding and their aggregation. However, a complete understanding of this issue still requires further research at the molecular and cellular level.

The first important factor that affects protein aggregation in neurodegenerative diseases is the protein structure. In particular, primary and secondary structures are critical factors affecting the physical and chemical properties of proteins/peptides and their three-dimensional conformation (tertiary structure). In the primary structure, the position and number of different characteristic amino acid residues can accelerate or slow down the aggregation process. In general, the number of hydrophobic amino acids in proteins is proportional to the aggregation tendency [6]. Mutations are another factor that affects the protein structure and are considered to have an impact on the aggregation process by changing protein solubility and stability [10].

Neurodegenerative diseases are associated with the occurrence of amyloid plaques, which are specific for each disease. AD pathogenesis is considered to be related to tau protein. Tau protein in its native state has important physiological functions, such as microtubule stabilization [11,12]. To perform their functions, tau proteins must be phosphorylated at a normal, physiological level. The hyperphosphorylation causes the loss of their biological activity due to conformational changes and abnormal aggregation of the tau protein [13].

Amyloid β (Aβ) is a small peptide with a mass of 4–4.4 kDa [14]. It is the main component of amyloid deposits found in AD, mainly in the cortex, hippocampus, forebrain, and brain stem [8]. The native Aβ occurs in neurons, astrocytes, neuroblastoma cells, hepatoma cells, fibroblasts, and platelets [15]. Its functions are associated with inflammatory and antioxidant activity. Aβ also affects the regulation of cholesterol transport and the activation of the kinase enzyme. Native Aβ protein has mainly disordered structure but several research reveals the presence of local regions displaying a secondary structure [16,17]. The relative β-sheet content increases with the ongoing aggregation associated with the AD [18].

α-synuclein (α-syn) is associated with Parkinson’s disease. The mass of the α-synuclein monomer is ca.14 kDa. The name “α-synuclein” comes from its synaptic and nuclear location. α-Syn regulates dopamine neurotransmission by modulation of vesicular dopamine storage. This protein interacts with tubulin. It also has a molecular protective activity in the folding of SNARE (soluble N-ethylmaleimide-sensitive-factor attachment protein receptor) proteins [19]. Native α-syn is soluble in the cytoplasm in contrast to its abnormal aggregates, which form Lewy bodies. The secondary structure of the α-syn aggregates is complex and consists of β-sheet, β-turns, α-helics/disordered conformations [20].

One of polyglutamine-containing, neurodegenerative proteins is a Huntington protein (Htt). It has a large mass of 350 kDa and consists of 3144 amino acids. Native protein is highly expressed in the peripheral tissues and the brain. This protein is involved in endocytosis, vesicle trafficking, cellular signal transduction, and membrane recycling. In the brain, abnormally aggregated Htt protein damages cells and it is toxic, forming pathological aggregates [21,22].

Monitoring the aggregation process with spectroscopic methods involves tracking the changes in the protein secondary structure. To correctly interpret the progress of the amyloid aggregation based on the spectroscopic data, it is necessary to understand the course of the aggregation process itself. The abnormal protein aggregation is considered to be related to protein misfolding leading to the exposure of hydrophobic amino acid residues or with the change of the protein net charge [8,23]. The exact aggregation pathway of amyloid proteins depends on the secondary structure of aggregating molecules. However, the main principles of the amyloid aggregation process can be simplified to a three-step fibril formation process consisting of the lag-phase, the elongation phase and the plateau phase (Figure 1) [8,24,25]. Within the lag phase, also called a nucleation phase, the soluble oligomeric intermediates are formed. The elongation phase (fibril growth phase) includes the formation of other “aggregation-prone” intermediates known as protofibrils. A self-template growth mechanism leads to their rapid growth until the formation of insoluble mature fibrils [8,26]. The plateau stationary phase is reached when the content of mature fibrils is at a constant maximum level. There are also theories describing the fragmentation of fibrils during the aggregation process changing the kinetics of mature fibril formation [26,27].

Mature amyloid fibrils, regardless of the peptide/protein they are made of, possess universal characteristic structural features. The morphology of fibrils is represented by a unique cross-β structure consisting of hydrogen-bonded β-sheets arranged perpendicularly along the fibril axis. The fibril diameter is usually 6–13 nm and is formed from 2–8 protofilaments twisted around each other [8,24,26].

The amyloid aggregation process and its subsequent steps can be followed with vibrational spectroscopy techniques. Spectra of proteins have already been described well. Individual bands are attributed to vibrations of various chemical groups present in the peptide chain. Amide bands A and B as well as amide bands I-VII (wavenumbers and assigned vibrations are shown in Table 1) can be observed in the Raman and infrared spectra of peptides and proteins. Molecular spectroscopy techniques are sensitive to the changes of the protein secondary structure. Each of the intermediate species of amyloid aggregation, as well as fibrils, are represented by a different ratio of individual secondary structures. In general, the *β*-sheet content increases with the progress of the aggregation process [28,29]. Celej et al. [28] registered the decrease in random coil and/or helical structure content (absorbance at 1660–1650 cm^−1^), for *α*-synuclein aggregation, from ca. 60% in monomers to 52% in oligomers and 27% in fibrils while the β-sheet contribution increased from 8% for monomers, 26% in oligomers up to 51% in fibrils. In this case, the *β*-turn content (at 1670 cm^−1^) was ca. 22% for oligomers and fibrils. The different ratio of secondary structures for fibrils formed from various peptide variants is considered to be a strong premise for different aggregation pathways [30,31,32].

This review is devoted to the description and organization of all molecular spectroscopic markers of abnormal protein aggregation. Due to comprehensive presentation of the infrared, Raman TERS, SERS, nanoFTIR and AFM-IR spectral changes related to the secondary structure modifications of neurodegenerative proteins and model peptides, researchers can easily verify their results, find an interpretation of the observed spectral changes and compare them with results obtained so far for various amyloids systems with all known molecular spectroscopic techniques. In addition, the usefulness of multivariate data analysis in extraction of spectroscopic markers of the amyloids aggregation is presented.

## 2. Infrared Spectroscopy Studies of Abnormal Aggregation of Proteins/Peptides

Infrared spectroscopy (IR), along with Raman spectroscopy, is one of the vibration spectroscopy techniques. The leading phenomenon underlying IR spectroscopy is the absorption of electromagnetic radiation in the infrared range by molecules, exciting vibrations and oscillations of the functional groups occurring in the studied analyte [35,36]. Each chemical bond undergoes various types of vibrations (i.e., stretching, bending, twisting, rocking or wagging motions), whose energies correspond to the IR region including far, mid, and near infrared. IR spectroscopy is selective for vibrations leading to the change of the molecule dipole moment and these vibrations are detected in the IR spectrum [35,37]. Due to the chemical sensitivity, as well as the susceptibility to intra- and intermolecular effects (which affects the vibration frequency and the bonds polarity), IR spectroscopy is very common in the science of biological systems. Although the IR spectra of biomolecules are very complex and consist of several overlapping bands (resulting in a loss of some information), they are still extremely useful for tracking structural changes of biological molecules. For example, it is possible to follow each process that affects the molecule geometry. In the case of proteins, IR spectroscopy makes it possible to analyze modifications in the protein secondary structure which, according to Barth [35], is probably the most popular application of this technique. The IR spectrum of proteins or peptides contains a few characteristic bands carrying the information about the secondary structure, known as the amide I (1700–1600 cm^−1^), amide II (∼1550 cm^−1^) and amide III (1400–1200 cm^−1^) bands. The amide I region is especially sensitive to the structure of the protein backbone and is the most useful in neurodegenerative peptide aggregation research.

For studying biological systems, in addition to the transmission FTIR spectroscopy (Fourier transform infrared spectroscopy), the attenuated total reflectance (ATR) technique seems to be advantageous. In an ATR, a sample is placed on a crystal with a high refractive index. The light is reflected once or several times on the crystal–sample interface, which helps to increase the sensitivity allowing to measure thin films. The evanescent wave, perpendicularly oriented to the crystal–sample interface, arises and penetrates into the sample, where it can be absorbed. Due to this phenomenon, the light reaching the detector contains information about the sample structure [35,38]. ATR is especially helpful in biological molecules science, where researchers are struggling with small amounts of the studied material (even less than 100 ng) and low sample concentrations (in the range of μM) [26].

Historically, the interpretation of amyloid aggregate (fibril) structures based on IR spectra was confusing. According to Sarroukh et al. [26], FTIR spectroscopy has been used to study fibril structures since the early 70ties. However, as was pointed out in the mentioned review, in the earliest articles describing the infrared spectra of fibrils, the major components of the amide I band were misinterpreted. The first paper which helped with the proper assignment of the amide I region peaks on the FTIR spectra of fibrils was released in 2000 by Bouchard et al. [39]. Tracking the insulin aggregation process over time (at pH 2.3, 70 °C), it was shown that the high frequency component at 1690 cm^−1^ appears in the spectrum only for short incubation times and vanishes after 18 h of incubation. It became clear that the ca. 1690 cm^−1^ amyloid IR spectrum component can be assigned to structures other than fibrils with the cross-β structure present in the sample. In general, for amyloid peptide aggregation, the spectroscopic marker of the aggregation process and mature fibril formation is the presence of a ca. 1630 cm^−1^ strong peak in the amide I region attributed to parallel β-sheets. At the same time, the approximately 5-fold weaker peak at ca. 1695–1685 cm^−1^ characteristic for antiparallel β-sheets is not present in the spectrum. The simultaneous occurrence of two components at ca. 1695–1685/1630–1610 cm^−1^ attributed to antiparallel β-sheets is visible in the IR spectrum when the aggregation process is in progress, at its initial stages, and oligomers are still present in the sample [40]. The IR marker spectroscopic bands of neurodegenerative peptide aggregation are presented in Table 2.

The aggregation of amyloid β (Aβ) peptides using IR spectroscopy has been extensively studied, especially in the past two decades. IR characteristic peaks occurring in the amide I region for the major component of amyloid β deposits, amyloid β_1-42_ [40,41] and amyloid β_1-40_ peptide [42,43,44] have already been described. To better understand the Aβ aggregation process, Juszczyk et al. [31] followed the aggregation of synthetic 11–28 fragment of the Aβ peptide with FTIR spectroscopy. This sequence is considered to be responsible for Aβ aggregation. The authors also investigated the aggregation of Aβ 11–28 fragments after the introduction of five different mutations (E22K, A21G, D23N, E22G and E22Q) in 21–23 position. The process was studied in HFIP–D_2_O mixtures with increasing water content.

The cytotoxicity of neurodegenerative proteins is considered to be strongly correlated with the secondary structure of entities occurring at subsequent stages of the amyloid aggregation process [8,24,28,45,46]. To confirm stronger cytotoxicity of oligomers with antiparallel β-sheet conformation over fibrils, Sandberg et al. [32] incorporated the double-cysteine mutation into the amyloid β_1-42_ and amyloid β_1-40_ (called Aβ_42_CC and Aβ_40_CC, respectively), which prevents mature fibril formation by stabilizing the oligomer structure with an additional intramolecular disulfide bridge. Both mutants create stable oligomeric molecules but within a different aggregation pathway. Aβ_42_CC oligomers/protofibrils turned out to be ca. 50 times more efficient in apoptosis induction than Aβ_1-42_ monomers or mature fibrils.

The aggregation of *α*-synuclein, the 140-amino-acid protein most abundant in Lewy bodies occurring as fibrillar intraneuronal inclusions in Parkinson’s Disease (PD), has also been studied with IR spectroscopy [28,30,47,48]. In general, infrared spectral markers of *α*-synuclein aggregation are similar to those found for amyloid β (see Table 2). An interesting work concerning *α*-synuclein aggregation was released lately by Ruggeri et al. [30]. Due to medical evidence of missense mutations in the *SCNA* gene encoding *α*-synuclein involvement in the PD pathogenesis, the aggregation of a wild type *α*-synuclein, as well as its variants with the one amino acid replacement in the protein sequence, has been studied. Results showed that amyloid fibrils formed by different variants of *α*-synuclein were varying in the percentage ratio of secondary structure content. This was due to alternative mechanisms of the aggregation pathway for studied protein variants. Another recent work concerning *α*-synuclein aggregation describes the influence of the ionic strength on the β-sheets orientation in fibrils studied with 1D- and 2D-IR spectroscopy [49]. 2D-IR spectroscopy provides information about the specific residues of interest and is sensitive to more ordered structures in general [49,50]. The β-sheet arrangement in fibrils turned out to be correlated with salt concentration during fibrilization. *α*-synuclein aggregation in low ionic strength conditions (NaCl concentration  ≤  25  mM) results in parallel β-sheet orientation in fibrils, while the fibrilization upon high salt concentration, including physiological conditions, contributes to the antiparallel β-sheet arrangement.

Natalello et al. [51] studied the aggregation pathway of prion peptide PrP_82–146_ characteristic for another neurodegenerative amyloid disease, called Gerstmann–Sträussler–Scheinker syndrome (GSS). The major components occurring in brain amyloid plaques are the PrP peptide fragments consisting of 81–82 to 144–153 amino acids. The FTIR spectra of the 82–146 PrP fragment at the initial stages of aggregation revealed two components, typical for oligomers displaying antiparallel structures: a low-frequency band around 1623 cm^−1^ simultaneously with a high-frequency band around 1690 cm^−1^. At the final stages of aggregation, when the sample was rich in the cross-β structure fibrils, only the 1626 cm^−1^ peak was determined.

Transthyretin (TTR), a biologically relevant protein occurring in human plasma, serves as thyroxin carrier, or a retinol binding molecule. The misfolding or abnormal aggregation of TTR leads to amyloidosis such as senile systemic amyloidosis (SSA), familial amyloid polyneuropathy (FAP), and familial amyloid cardiomyopathy (FAC) [52]. TTR fibril formation was studied by Cordeiro et al. [52] and earlier by Zandomeneghi et al. [53]. The spectral maker for TTR fibril formation is the band at 1625 cm^−1^.

In addition to neurodegenerative peptide aggregation, the secondary structure of amyloid fibrils of other non-neurodegenerative peptides has also been extensively studied. The IR marker bands of abnormal protein aggregation have been determined for β2-microglobulin (β2m), whose deposits occurred in dialysis-related amyloidosis within the musculoskeletal system [54], a prion-forming 218–289 domain of HET protein [55], wild-type human lysozyme [29] and hen egg white lysozyme (HEWL) [56] (see Table 2). These proteins have been studied as great models of amyloid aggregation. Zurdo and co-workers [57] characterized the fibrils formed by the SH3 domain of the α-subunit of bovine phosphatidylinositol-3′-kinase. The FTIR spectra were collected for amorphous aggregates as well as for fibrils (created at low pH) exposed to pepsin to receive a sample containing only a fibrillar structure with characteristic low frequency band for parallel β-sheets (1618 cm^−1^). Another amyloid-like structure formed from a prion-like protein, Sup35, derived from yeast, was studied by Balbirnie et al. [58]. FTIR spectra of aggregated protein crystals showed the 1633 cm^−1^ parallel β-sheet band. The aggregation kinetics of human islet amyloid polypeptide (hIAPP) was studied using 2D infrared spectroscopy combined with site-specific isotope labeling [59]. This methodology allowed to follow the intensity growth of the 1617 cm^−1^ peak related to the increase of β-sheet content upon aggregation and fibril formation. Ami et al. [60] incorporated the FTIR microscopy to study aggregates of amyloidogenic immunoglobulin light chains (LCs) occurring in the light chain (AL) amyloidosis pathology. The applied methodology involved the FTIR in situ studies of unfixed tissues (hear and subcutaneous abdominal fat) derived from AL amyloidosis affected patients as well as the research of in vitro aggregated peptide (derived from a patient). The infrared β-sheet signature characteristic for amyloid aggregation was possible to detect in situ in the spectra of tissues.

### 2.1. Infrared Spectroscopy at the Nano Scale in Studies of Abnormal Proteins/Peptide Aggregation

Due to the diffraction limit, the resolution of conventional IR spectroscopy does not make it possible to track changes concerning single molecules. The signal reaching the detector contains bulk information, averaged over many molecules of the studied sample. To overcome this limitation and follow the IR absorbance spectra at the single molecule level, the novel nano-FTIR and AFM-IR techniques have been implemented.

#### 2.1.1. Nano-FTIR In Studies of Abnormal Proteins/Peptide Aggregation

The nano-FTIR technique makes it possible to achieve simultaneous infrared chemical and topographic characteristics of a sample at nanoscale resolution. It became possible due to the invention of scattering-type scanning near-field optical microscopy (s-SNOM), which is a unique combination of atomic force microscopy (AFM) with optical imaging and IR spectroscopy [61]. Nano-FTIR combines the nanometric spatial resolution of AFM with the chemical sensitivity of IR spectroscopy. The local probing of molecular vibrations involves the incident light scattering at an AFM tip apex (Figure 2). The electric charge accumulated on the probe apex generates a localized electric field. When the AFM probe (usually metallic) is in contact with the sample, it is possible to detect the near-field interaction of infrared light with the sample due to the induction of a local evanescent field [61]. Depending on the measurement purposes, the nano-FTIR technical setup provides a broadband mid-infrared laser (fiber laser plus difference frequency generator) or tunable single line laser (quantum cascade laser, QCL). Both of these light sources enable chemical imaging, the broadband laser makes it possible to obtain the hyperspectral maps and the QCL provides maps at the fixed wavelength. When the interferometer is operating as a Fourier transform spectrometer, it is possible to collect single FTIR spectra at points of interest on the sample, selected precisely based on the AFM image [61].

Nano-FTIR was employed to study amyloid aggregates by Amenabar et al. [61]. In this work, it was shown for the first time that nano-FTIR can be used to study the secondary structure of individual proteins. Amenabar and co-workers explored the conformation of single insulin fibrils. The nano-FTIR spectra of insulin fibrils (Figure 3) revealed the bands characteristic for β-sheets at 1639 cm^−1^, α-helical structures at 1671 cm^−1^ and the band at 1697 cm^−1^ whose authors attributed to β-turns or antiparallel β-sheet. The weak band at 1609 cm^-1^ was assigned to amino acid side chains. The images at single wavelengths provided by QCL laser (Figure 3c) as well as single broadband spectra (Figure 3d) confirmed the presence of α-helices in 3-nm and 9-nm thick fibrils.

#### 2.1.2. Infrared Spectroscopy Combined with Atomic Force Microscopy (AFM-IR) in Studies of Abnormal Proteins/Peptide Aggregation

Infrared nanospectroscopy, also called Photothermally Induced Resonance (PTIR) or infrared spectroscopy coupled with atomic force microscopy (AFM-IR), enables the measurement of a local infrared light absorption [62]. Thanks to the use of tunable infrared lasers it is possible to obtain single spectra or maps showing the spatial distribution of selected molecules and their functional groups. Conventional spectroscopic methods, such as IR or Raman spectroscopy, are limited by the diffraction criterion. The spatial resolution of these techniques depends on the wavelength and is in the range of 5–10 µm [63]. Therefore, they cannot be applied in imaging of micro- or nanometric biological objects. The PTIR method has been first demonstrated in 2005, by prof. A. Dazzi from Université Paris Sud [62,63,64]. This technique is based on the detection of infrared radiation absorption using atomic force microscopy (AFM). Transient and local sample extension are detected as a change of AFM cantilever deflection (Figure 2). Photothermally induced signal increases when the laser is tuned to frequencies which are absorbed by the sample. The limitation of AFM-IR spatial resolution is determined by scanning and sampling rates and is related to several factors such as the tip apex size and thermomechanical properties of the investigated sample [62,63,64].

An application of AFM-IR in the structural organization of individual fibril aggregates is of central importance for several scientific groups. Henry and co-workers studied the amyloid β (Aβ_1-42_) peptide and its G37C mutant at the nanoscale [65]. They used AFM-IR to examine the interaction of lipids such as 1-palmitoyl-2-oleoylphosphatidylcholine, sphingomyelin, or cholesterol with amyloid aggregates. It was discovered that aggregation of amyloid β_1-42_ and its mutant essentially changes the structural organization of the fibrils in the presence of the investigated phospholipids. Ruggeri et al. applied AFM-IR to investigate single aggregates of a highly fibrillogenic domain of ataxin-3 called “Josephine” and followed its whole aggregation pathway [66]. They observed that already at the first aggregation, abnormal folding of Josephine was formed. It was possible to detect conformational transitions starting from native oligomers to misfolded oligomers and finally to mature amyloid fibrils. Ruggeri and co-workers also studied Exon1 aggregation which is the first exon of Hungtinton protein [67]. They determined the influence of the polyQ content and the Nt17 domain occurrence on the biophysical features of Exon1 fibrils. In the presence of the Nt17 domain, the IR spectra of fibrils revealed the peak characteristic for β-turn rich secondary structure at 1684 cm^−1^ while the lack of this domain changed the secondary structure of fibrils which displayed antiparallel β-sheet structure. Rizevsky et al. [68] studied insulin fibrils using AFM-IR. In the following work, the formation of two different types of fibers was described. This phenomenon is associated with the insulin aggregation pathway. The first fibril type had an β-sheet-rich secondary structure, whereas the second polymorph revealed mainly an unordered secondary structure. Galante and co-workers [69] studied the influence of the most toxic, pyroglutamylated isoform of amyloid β (AβpE3-42) on the biophysical features and biological activity of amyloid β_1-42_ [70,71]. It was confirmed that the mixture of ApE3-42/A1-42 with the 5% content of AβpE3-42 isoform was the most toxic. Ramer et al. [72] applied AFM-IR for the first time to investigate the chemical structure of amyloid aggregates at the nanoscale in water. They demonstrated that it was possible to collect PTIR spectra in liquid with a high signal-to-noise ratio. They studied aggregates of diphenylalanine (FF, Figure 4b), which is the core amyloid β peptide, and its tert-butoxycarbonyl derivative (Boc-FF, Figure 4a). It was confirmed that the detection of differences in the studied fibrils is possible for measurements conducted in liquids. Boc-FF appeared to have a more helical conformation in comparison to FF. These results are promising, and make it possible to state that AFM-IR can be widely used for very composed aggregation systems in liquids. AFM-IR marker bands related with the neurodegenerative peptide aggregation are presented in Table 3.

## 3. Raman Spectroscopy in Studies of Abnormal Proteins/Peptide Aggregation

Raman spectroscopy is a non-destructive analytical technique that makes it possible to obtain information about the chemical structure and composition of the investigated sample. The basic phenomenon used in this method is an inelastic light scattering, which contains information about the energies of the vibrational states [77] (the light scattering mechanism is shown in Figure 5). The energy difference between incident radiation and scattered radiation is corresponding to the energy levels of vibrational modes in functional groups of the investigated molecules. This process causes the excitation of characteristic vibrations, which are observed in the spectrum as specific bands [77,78].

In respect to these relationships there is a selection rule, and according to this, the polarity of the molecule must change. Inelastic scattering, also called Raman scattering, is very rare. Approximately only one photon per 10^10^ is scattered inelastically [78]. For this reason, various methods to amplify the signal are proposed, such as surface and tip-enchanted Raman spectroscopy, which are described in the next paragraphs of this article. Raman spectroscopy is widely used in the study of various types of biological material [79], and its use in medical diagnostics is being considered [80,81].

Deep UV Resonance Raman (DUVRR) Spectroscopy is a variant of the Raman spectroscopy very commonly used in the research of protein aggregation. In contrast to Non-resonance or Normal Raman (NR) Spectroscopy, the radiation used in the experiment (in this case UV spectral range) transfers the electron to a higher energy state, which causes resonant amplification of the scattered light signal (Figure 6). Thus, it becomes possible to observe bands that are less visible in the NR method [82,83].

As a result of the aggregation process, structural transformations occur in proteins, which manifest as characteristic changes in the Raman spectra (Table 4). An increase in the intensity of the bands from amide I and II is observed, due to the formation of β-sheets. This process was observed in various forms, depending on the proteins tested [84,85,86]. Kurouski et al. observed an increase in the intensity of amide I and II bands in the formation of insulin fibrils [86], while in the case of hen egg white lysozyme (HEWL) studied by Rosario-Alomar et al. only the change of amide II was visible [85]. In addition, a shift of amide I and II bands towards higher energies was observed [85,86]. The second characteristic marker that almost always appears is an increase in the intensity of the C_α_-H band at 1390 cm^−1^, which is explained by the decay of α-helix into a disordered structure [85,87]. In addition, there are changes in the intensity of the complex amide III band. Xu et al. described an increase in intensity in the range of 1270–1320 cm^−1^ and a decrease in the range of 1230–1270 cm^−1^ [88].

In the spectra of aggregated proteins, there is often a change in the band’s intensity from individual amino acids. In particular, a characteristic peak for phenyloalanine (1004 cm^−1^) is often described as one of the spectroscopic markers of fibrillation [88]. In some cases, there is a significant decrease of Phe band intensity [87,88], which is explained by the change in the chemical environment of this amino acid. However, this change is not always observed [86].

Interesting changes in Raman spectra were detected for the disulfide band in the range of 450–550 cm^−1^. Kurouski and Lednev observed significant changes in the spectrum for apo-α-lactalbumin (LA) and 1-SS-carboxymethyl lactalbumin (1-SS-LA) [84]. In the spectra of native LA, two peaks were observed at 510 cm^-1^ and 530 cm^−1^ due to various secondary structures. In LA fibrils, the β-sheets structure is mainly predominant; therefore, in their spectra, one peak at 508 cm^−1^ was observed. In turn, 1-SS-LA showed a slightly more complex structure, but also in this case a significant change in the spectra related to secondary structure transition was visible. Similar studies were carried out by Rosario-Alomar et al. for HEWL. Along with ongoing fibrillation, two peaks at 507 cm^−1^ and 523 cm^−1^ merged into one at 490 cm^−1^ [85].

### 3.1. SERS In Studies of Abnormal Proteins/Peptide Aggregation

Surface-Enhanced Raman Spectroscopy (SERS) is based on the enhancement phenomenon in Raman scattering by the application of nanostructures consisting of noble metals, transition metals, or semiconductors. For all molecules adsorbed onto nanostructured metal surfaces inelastic light scattering is greatly enhanced (enhancement factor can be up to 10^10^) in comparison to free molecules [89]. It is considered that the SERS effect is a combination of two mechanisms: an electromagnetic field enhancement (EM) and chemical surface interactions. EM field enhancement is caused by the excitation of surface plasmons on the surface due to the interaction between the electrons in metal nanostructure with the incident electromagnetic radiation. The locally enhanced electromagnetic field on the nanoparticle surface strongly increases the intensity of Raman scattering, because the Raman scattering cross-section is proportional to the electromagnetic field. On the other hand, the Raman scattering cross-section also strongly depends on the polarizability of the investigated molecule. Chemical enhancement is attributed to a significant increase in the polarization of the molecules due to its absorption onto the metallic surface, which results in the charge transfer affecting significantly the polarizability. New energy states are created that can be excited with the laser beam. This Raman resonance enables electron transfer from the Fermi level of the metal to the lowest unoccupied molecular orbital of the molecule (LUMO) and from the highest occupied molecular orbital (HOMO) to the Fermi level of the metal. The chemical enhancement is less effective and strongly depends on the type of adsorbed molecules, while the electromagnetic enhancement is universal for all molecules [90,91].

During the last decade, SERS has become quite popular as one of the most sensitive analytical techniques in chemistry, material science and biotechnology. It provides information about conformational and structural changes in molecules at very low concentrations. However, protein investigation still remains challenging. First of all, SERS measurements suffer from low spectral reproducibility. The SERS enhancement factor is strongly determined by the distribution of nanoparticles. Variations in spectral characteristics are therefore induced by inconsistent aggregation and collocation of the nanoparticles. Thus, the uncontrolled enhancement of signal from proteins adsorbed on the SERS-active surface generates SERS spectra with low reproducibility. Moreover, proteins, as the intrinsic molecules, often form complex SERS patterns, which makes identifying characteristic Raman fingerprints difficult [92,93,94]. To overcome this limitation, it is important to ensure that protein binding to a SERS-active substrate is consistent and well understood. Constant enhancement and reproducibility of SERS substrate preparation are therefore of central importance.

Due to the fact that SERS technology provides information about the secondary structure, it allows for an understanding of properties and functionality of the abnormal protein aggregates at very low concentrations. SERS marker bands associated with the abnormal amyloid aggregation are summarized in Table 5. One of the first SERS studies on protein aggregation concerned amyloid β [95]. Choi et al. used a nanofluidic device with SERS active gold nanoparticles to determine and characterize various stages of amyloid β aggregation. To understand conformational changes induced by protein aggregation, SERS spectra were collected as a function of time and protein concentration. The analysis of the amide III band allowed to investigate the formation of amyloid aggregates, the decrease in α-helics and the increase in β-sheet at micromole to femtomole concentrations [96]. A similar study was conducted by a group from Texas University [97]. Further research related to the conformational changes of amyloid aggregates was provided. Bhowmik et al. used Surface-Enhanced Raman Spectroscopy with silver nanoparticles to determine the Aβ oligomers structure that spontaneously bind to lipid bilayer. The secondary structure information of individual residues was examined based on Raman isotope shifts. The authors determined that the structure of Aβ_40_ oligomers bound to the membrane, display β-turn in the 23-28 region, situated between antiparallel β-sheets [98]. Insulin aggregation has also been investigated. Kurouski et al. used surface-enhanced Raman scattering to study the kinetics of insulin aggregation followed via detection of the insulin prefibrillar oligomers secondary structure. Insulin at different stages of aggregation was mixed with 90 nm of Au nanoparticles. SERS analysis demonstrated a more than two-fold increase in the number of insulin oligomers after the first hour of incubation. Further protein incubation resulted in a slow decrease in oligomers amount. The observed decrease in the quantity of prefibrillar species was linked to their conversion into fibrils [99]. Li et al. discovered that bromophenol blue (BPB) potentially inhibits the insulin fibrillation process [100]. Karabelli et al. used the electrochemical SERS (EC-SERS) method to study interaction between intermediates occurring at various stages of insulin aggregation and biomimetic membrane. It was observed that protofibrils and oligomers evoked significant membrane perturbation, whereas the native protein seems to have a protective role. The authors presented one of the first applications of the EC-SERS technique to investigate protein aggregation intermediates–biomembrane interactions [101]. Yu et al. discovered a method that makes it possible to distinguish Aβ_40_ and Aβ_42_ peptides easily. SERS combined with principal component analysis (PCA) revealed changes in peptide conformation and self-association [102]. Undeniably, SERS has great potential in peptide/protein studies, continuous work on increasing spectrum repeatability will certainly provide a deeper understanding of protein aggregation.

### 3.2. TERS In Studies of Abnormal Proteins/Peptide Aggregation

Tip-enhanced Raman Spectroscopy (TERS) takes the advantage of the nanometric spatial resolution from Scanning Probe Microscopy (SPM) including Atomic Force Microscopy (AFM) or Scanning Tunnelling Microscopy (STM), and the chemical selectivity of Raman spectroscopy. Therefore, it provides information about the molecular structure and composition with nanometric spatial resolution [103]. The excitation of surface plasmons in a metal nanostructure, which can be deposited on the apex of an AFM tip (AFM-TERS) or in STM tip apex itself (STM-TERS), modifies the electromagnetic field of incident laser light [103,104]. The Raman scattering cross-section is proportional to this electromagnetic field, and due to the enhancement, it can be dramatically improved. What is more, the generated electromagnetic field is highly localized; therefore, spectra are acquired from the small amount of sample located directly under the tip [103,104]. TERS tip can be understood as a nanoantenna. It converts the electric field of the Raman laser into highly spatially confined energy [105], breaking the Rayleigh’s diffraction limit, and improving the spatial resolution to a few nanometers. On the other hand, TERS probe converts the near field of the sample to a far field, accessible to a microscopic objective, increasing the sensitivity to the single molecule level [106,107,108]. In recent years, TERS has mainly been applied for samples with a high natural Raman scattering cross-section including carbon nanotubes and dyes [106,107,108]. The main limitation of TERS is actually related to its high sensitivity. The Raman scattering cross-section of carbonaceous substances and contaminations is exceptionally high [109]. Even thin traces of carbon contaminations can be easily detected as rapidly fluctuating and sharp peaks, which averaged over many spectra result in the broad D- (1360 cm^−1^) and G-bands (1580 cm^−1^) [109]. A very strong electromagnetic field, however, could damage delicate biological samples and cause their photochemical/thermal decomposition.

TERS delivers highly resolved topographic information and provides a chemical composition of the investigated sample. Therefore, this analytical technique is potentially ideal for studying protein secondary structure and its modification upon the fibrillation process. However, TER marker bands required for secondary structure identification such as amides I and III or C-H/N-H motions, allowing for conformation detection based on spectral positions, are mostly not well resolved. In particular, the most informative amide I band is absent in TER data. This problem has been broadly discussed [110,111,112], and finally it was proved that peptide backbone bonds may dissociate in the laser hot spot, which causes the lack of the amide I band. An application of the mid measurement condition (reduced laser exposition time and laser power) prevents the decomposition of the peptide backbone improving the spectra quality.

Due to the TERS limitations mentioned in the previous paragraphs, measurements of biological samples including proteins are still very limited. However, the results obtained so far are undoubtedly very valuable and should be presented in this review. For clarity, TER marker bands related to the abnormal amyloid aggregation are presented in Table 6. One of the first neurodegenerative peptides characterized with TERS was β2-microglobulin [113]. Several polypeptides were intensively studied by the group of prof. Deckert from Jena University. Deckert and co-workers applied TERS to analyze the surface molecular structure of human islet amyloid polypeptide fibrils, proving that the highly heterogeneous fibril surface mainly contains α-helical or unordered structures, in contrast to the fibril core, which is built of β-sheets [114]. Insulin fibrils were also investigated. Kurouski et al. probed the chemical composition and the secondary structure of insulin fibrils [115], demonstrating the high capability of TERS in characterization of heterogeneous fibrils. Insulin was also applied to test the efficiency of several aggregation inhibitors including: benzonitrile, dimethylsulfoxide, quercetin, and β-carotene [116]. Analysis of the shape of the amide I band in the TERS spectra makes it possible to detect the influence of these inhibitors on fibrils secondary structure [116]. It was proven that all insulin aggregates showed similar morphology but various inhibition results in their different β-sheet structure content, suggesting serious modification in the aggregation pathway. Specifically, the authors suggested that prevention of the β-sheet stacking of the peptide chains played a major role in the aggregation inhibition [116]. Additionally, the group from Jena identified the secondary structure of insulin aggregates via the amide III band and what is more, performed successful detection of hydrophobic and hydrophilic domains on the surface [117]. Bonhommeau et al. characterized Aβ_1–42_ and its two synthetic, less toxic mutants at the nanoscale. Based on the examination of amides I and III bands, detection of fibers organized in parallel β-sheets and in anti-parallel β-sheets was possible.

In previous studies, single-point TER spectra were mainly acquired at selected locations. However, recently, hyperspectral TER mapping, which consists of a full TERS spectrum in each pixel of the image, was performed. Paulite et al. successfully applied STM-TERS to map Aβ_1-40_ through integration of the phenylalanine ring breathing mode [119]. The highly reproducible and intense peak from phenylalanine at 1004 cm^−1^ dominates each STM-TER spectrum; however, amides were not well resolved. Recently, AFM-TERS was applied for nano-spectroscopic hyperspectral mapping of amyloid β_1–42_. An application of AFM-TERS allowed for localization of β-sheet and unstructured coil conformation distribution in oligomers and along fibrils and protofibrils [118]. These studies confirmed the high capability of TERS in the mapping of amyloids, and what is more, they allowed for direct monitoring of aggregation pathways by following the distribution of the secondary structure in individual aggregates: oligomers, protofilaments, and fibrils. Figure 7 demonstrates the AFM image of A β_1–42_ deposited on a gold surface. The zoomed-in area is partially overlapped by the TERS map in the central part. Two types of TER spectra, presented below the maps, were acquired in the area of the individual fibril. The distribution of each of those spectra is superimposed on the zoomed AFM topography. The main difference between the two types of spectra acquired from the fibril is a significant shift of the amide III band from 1260 cm^−1^ (green spectrum) to 1250 cm^−1^ (blue spectrum). The position of the amide III band indicates the particular peptide secondary structure: turns/random coil and β-sheet conformation, respectively. The distribution of these conformations along a single fibril is presented in the zoomed-in area. Another characteristic marker of β–sheet structure, which can be observed in the blue spectrum, is a high intensity of the Cα- H/N-H bending mode at 1364 cm^−1^ (Figure 7). Despite the absence/low intensity of the amide I band in the studies described above, it was confirmed that TERS is an efficient tool for investigation of the amyloids secondary structure and for direct verification of the aggregation pathway [118].

## 4. Multivariate Data Analysis in Studies of Abnormal Proteins/Peptide Aggregation

The main purpose of performing a multidimensional statistical analysis (MSA) on spectroscopic data is to prove that measured spectral differences exist according to different populations and this information is statistically significant. The subject of considerations in multidimensional statistical analysis is a large set of data describing the studied phenomenon by means of vectors. MSA methods make it possible to organize this data, establish relationships between variables, as well as to estimate parameters and verify hypotheses of interest. Classical multivariate statistical analysis was developed assuming that the observable random vectors have normal distributions. However, in recent decades, there has been a rapid development of nonparametric multidimensional statistical analysis. Principal Component Analysis (PCA) and Hierarchical Cluster Analysis (HCA) are the most popular. However, in contrast to the PCA method, the use of HCA is not limited to all-numeric data. In the case of the spectroscopic methods with the single-molecule sensitivity as SERS, TERS, AFM-IR or nano-FTIR, the details obtained by performing the MSA analysis cannot be immediately apparent from the separate analysis of peak intensities. The different approach is commonly used in Raman or IR techniques due to the greater signal-to-noise ratio in those methods.

Many techniques have been developed to simplify data analysis, especially in the case of difficulties with interpretation of large datasets. Principal component analysis (PCA) is one of the oldest and most widely used techniques in this area. The main idea of PCA is to drastically reduce dataset dimensionality in an interpretable way without a significant loss of statistical information. The basic requirement in PCA is to successively maximize variance and finally to find new variables that are linear functions of those in the original dataset. Therefore, it finally leads to an eigenvalue/eigenvector problem solution. PCA was firstly incorporated for data analysis by Pearson [120] and Hotelling [121]. Although PCA needs no distributional assumptions, a multivariate normal distribution of the dataset establishment is commonly used. This method is an exploratory adaptive and can be used on various numerical data types. The PCA method makes it possible to reduce the noise and to detect the sub-groupings within the spectra. The most important results of PCA are the score plots and the loading plots. The score plots represent the degree of variability within the totality of spectra and the loading plots show which variables in the data set are responsible for the greatest degree of separation [122,123].

The overall goal of hierarchical cluster analysis (HCA) is similar to that of PCA, but the mathematical approach is different. The HCA is typically applied to determine how entities can be grouped into clusters, that exhibit high similarity within-group and low similarity to other groups. The result of HCA is usually presented as a dendrogram, which is a plot that shows the relationships between the samples in a tree form. In HCA, two main approaches (agglomerative and divisive) are used to resolve the grouping problem. In the agglomerative approach, each sample is initially considered as a cluster, and subsequently, pairs of clusters are merged. The divisive approach is contrary, the algorithm starts with one cluster including all of the samples, and recursive splits are performed. Clustering is achieved by the use of an appropriate metric of sample distance and linkage criterion among groups. The most commonly used, in the case of the datasets obtained from mentioned techniques, are Euclidean or Manhattan distance in combination with Ward’s method linkage. Clustering can be used for the identification of interesting patterns and distributions in the data structure. Therefore, it is a useful technique for discovering and extracting information that may previously have been unnoticed [124]. Cluster analysis was proposed in 1930; however, it has found applications across a wide variety of disciplines since the beginning of the early 1960s. One of the primary difficulties with the application of HCA for the extraction of significant information is the choice of the optimal number of clusters. This problem is still considered to be important despite several activities concerning this field. Although several criteria are available to assist in the proper selection of the appropriate number of clusters, the results depend on the researcher.

PCA was used to analyze spectroscopic data concerning amyloid aggregation for the first time by Ruggeri et al. [66]. The authors performed the PCA by means of the nonlinear iterative partial least squares algorithm with cross-validation and mean-centered data analysis of underivatized spectra and second derivatives calculated after an application of the Savitzky–Golay algorithm for smoothing. This analysis confirmed that shifts of the amide I (from 1655–1620 cm^−1^ to 1710–1680 cm^−1^) and the amide III bands (from 1380 to 1295 cm^−1^) are spectral markers of the Josephine domain of ataxin-3 protein aggregation. The secondary structure of the studied domain changed from the random coil/α-helical structure in native protein to β-turn/antiparallel β-sheet conformations in aggregated forms [66].

The HCA analysis was performed to obtain the averaged TERS spectra used as the reference for assigning the two conformations, turns/random coils, and β-sheet in the analysis of TER maps [99]. The correlations between each single spectrum from the acquired maps and the marker spectra determined by HCA were estimated with Pearson’s correlation coefficient. Calculations were performed using not the raw spectra, but rather their second derivatives to avoid any influence from the baseline. The same maps were also analyzed using principal component analysis for the statistical comparison. The multidimensional statistical analysis allowed to demonstrate the distribution of the turns/random coils and β-sheets secondary structures in individual amyloid β_1-42_ aggregates, therefore, Lipiec et al. verified spectroscopically several hypothetically proposed aggregation pathways [118].

The PCA performed on the SERS data allowed to distinguish amyloid β isoforms, Aβ_1–40_ associated with classical vascular and Aβ_1–42_, related to parenchymal vascular plaques in the Alzheimer’s Disease. Five different wavelengths were analyzed using the PCA method to maximize the between-group (data from each point) variation. During the mentioned analysis, the observation of a minimum of nine differentiable clusters within PCA spaces was implied, which corresponds to at least nine different assembly states in the fibril formation pathway. This approach finally proved to be extremely useful for determining structure-activity relationships [102].

## 5. Conclusions

In conclusion, the work presented in this review shows the enormous progress in the state of knowledge concerning amyloid aggregation. Spectroscopic studies of abnormal protein aggregation revealed characteristic marker bands related to specific secondary structure content occurring at successive steps of the aggregation process. The spectroscopic research carried out so far has made it possible to link the ratio fluctuations of individual secondary structures content in fibrils with the occurrence of alternative aggregation pathways. In particular, molecular nanospectroscopy has made it possible to study the distribution of particular conformations along individual amyloid fibrils that appeared to have a highly heterogeneous structure at the nanoscale. This tremendous progress allows one to go one step further and conclude about the specific aggregation mechanism of the studied system. Now, the greatest limitation of nanospectroscopy seems to be related to technical difficulties with transferring measurements into the liquid conditions. The drying process of biological components may influence their native structure. The efforts of nanospectroscopy community are focused on overcoming this limitation and the progress in this direction had already been presented in the work of Ramer et al. [72].

The research concerning abnormal protein aggregation has been focused so far on studying the aggregation of isolated/synthesized peptides and fibrils formed in vitro. That gave one the solid benchmark to study more complicated systems such as isolated amyloid plaques from neurodegenerative diseases or amyloidosis affected patients. Such works have been limited so far due to the complexity of such systems, but now it seems to be one of the future directions of spectroscopic studies concerning abnormal protein aggregation.

## Figures and Tables

**Figure 1 molecules-25-02498-f001:**
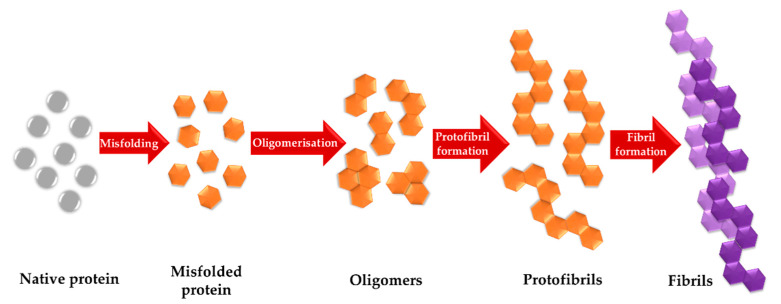
The schematic representation of successive species formation in amyloid aggregation process.

**Figure 2 molecules-25-02498-f002:**
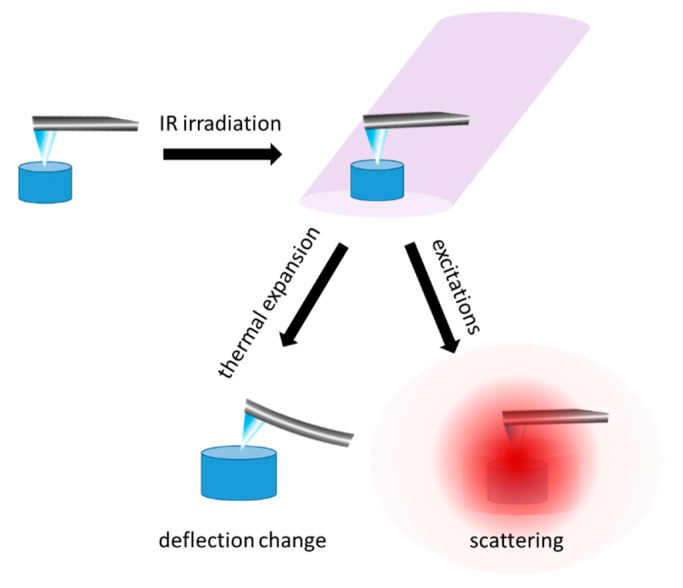
The schematic representation of infrared nanospectroscopy principles: thermal extension of a sample and a change of AFM cantilever deflection, as a consequence of the interaction with the incident IR light (AFM-IR) and scattering of the near infrared field from the sample at the metal/metalized probe apex (nano-FTIR).

**Figure 3 molecules-25-02498-f003:**
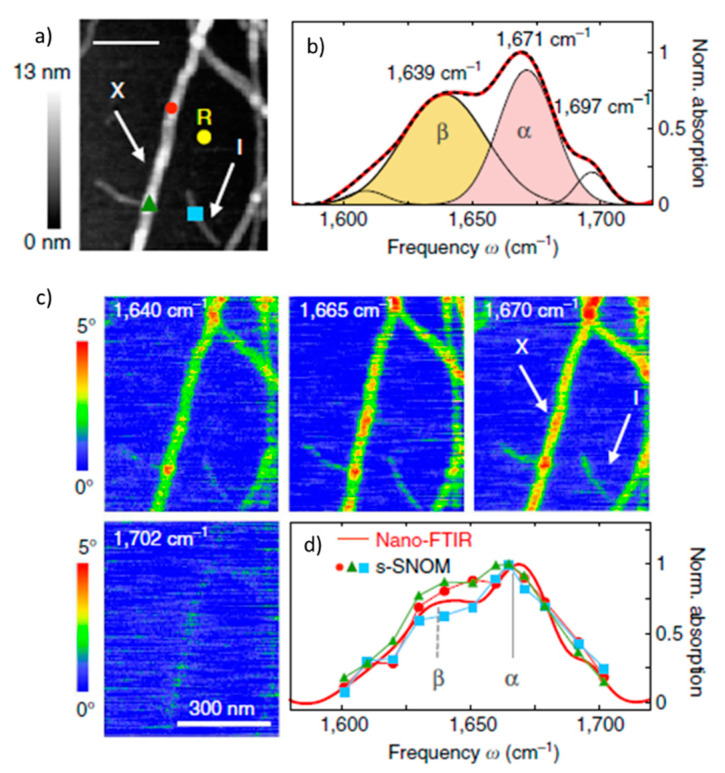
Infrared nanospectroscopy and nanoimaging of insulin fibrils. (**a**) AFM topography image of insulin fibrils, scale bar: 200 nm; (I) 3-nm thick type I fibril, (X) a 9-nm-thick fibril. (**b**) the nano-FTIR spectrum (red curve) based on five absorption bands. (**c**) s-SNOM phase images of the fibrils at fixed wavelengths, scale bar: 300 nm. (**d**) Single infrared absorption spectra at the positions marked in (a), topography image [61]. Images reproduced under CC BY license.

**Figure 4 molecules-25-02498-f004:**
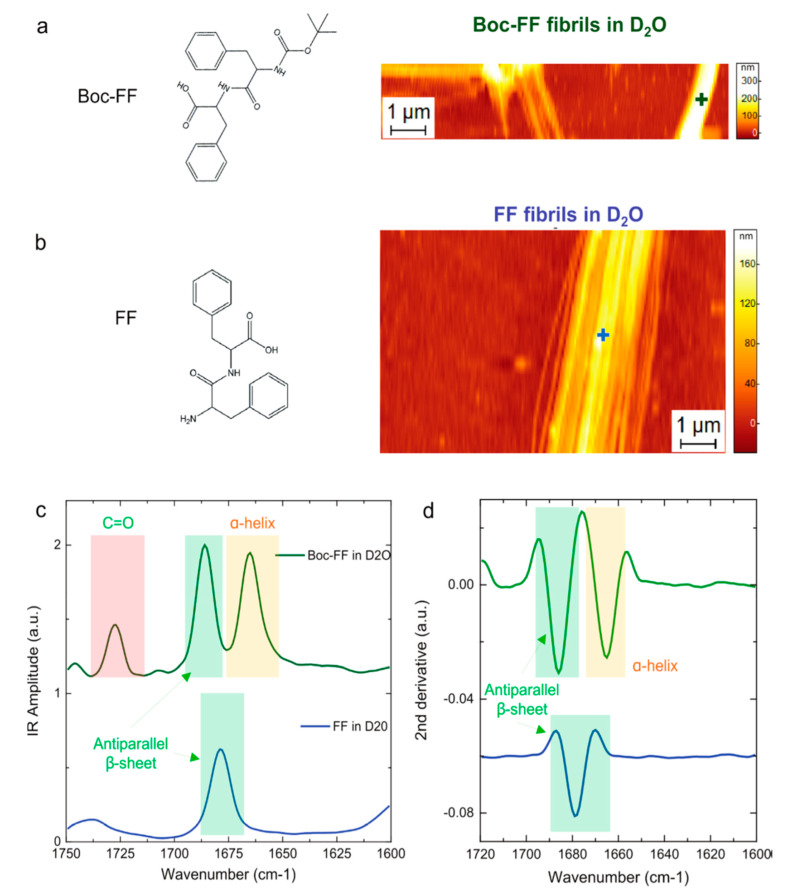
PTIR structural comparison of FF and Boc-FF fibrils in D_2_O. Chemical structure and AFM morphology maps of (**a**) Boc-FF and (**b**) FF fibrils. (**c**) Comparison of the fibrils averaged PTIR spectra in the amide I (green and yellow) and C=O stretching vibration (red) spectra ranges. (**d**) Comparison of the second derivatives spectra in the amide band I region [72].

**Figure 5 molecules-25-02498-f005:**
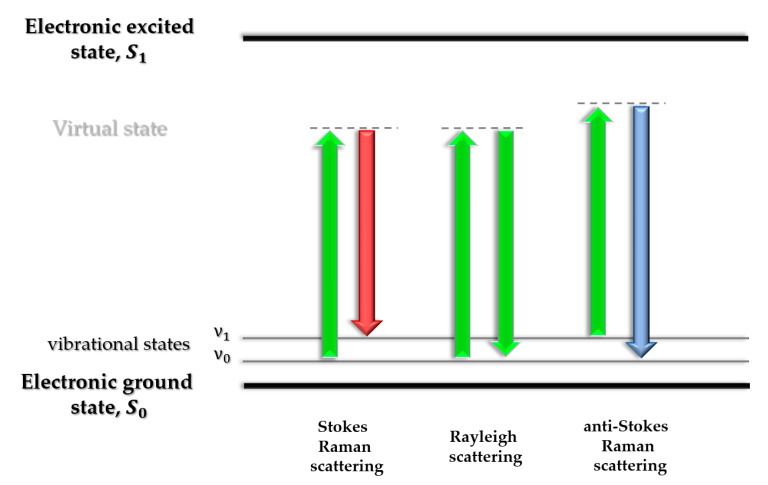
During non-resonant light scattering, three scenarios are possible: in the middle, elastic scattering, called Rayleigh scattering (without energy changes), on sides inelastic light scattering (causing energy changes), called Raman scattering including anti-Stokes (higher energy of scattered photon), and Stokes (lower energy) lines.

**Figure 6 molecules-25-02498-f006:**
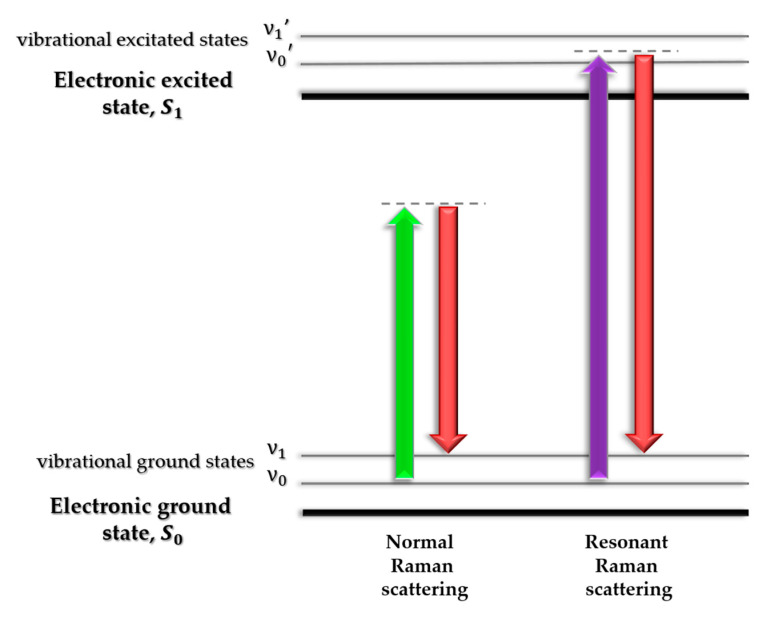
Comparison of NR and DUVRR spectroscopy. In NR, the electron is not transferred to a higher energy state, but only to a virtual state (lower energy than S_1_). In DUVRR, the radiation scattered on the sample excites the electron to the S_1_ state energy, causing the resonant signal amplification.

**Figure 7 molecules-25-02498-f007:**
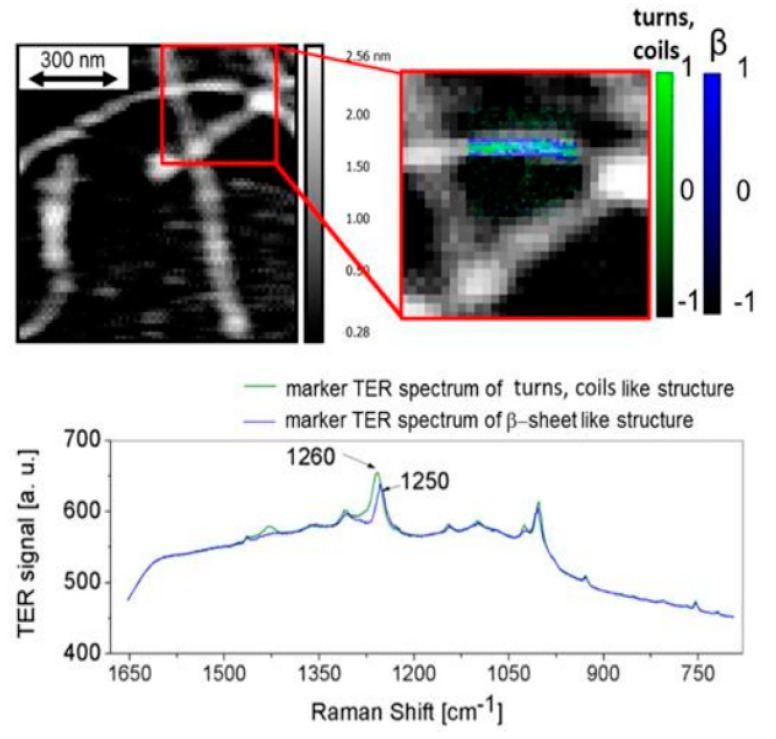
AFM-TER mapping of Aβ_1–42_ fixed on gold: AFM topography with overlapped TERS map in zoomed area showing the distribution of β-sheets (blue) and turns/unstructured coils (green) with corresponding averaged TERS spectra, characteristic for both conformations; adapted with permission from [118].

**Table 1 molecules-25-02498-t001:** Characteristic Raman and infrared bands for proteins [33,34]

Amide Bands	Wavenumber [cm^−1^]	Assigned Vibration ^1^
A	3500	ν(N-H)
B	3100	ν(N-H)
I	1700–1600	80% ν(C=O), 10% ν(C-N), 10% δ(N-H)
II	1580–1480	40% ν(C-N), 60% δ(N-H)
III	1300–1230	30% ν(C-N), 30% δ(N-H), 10% ν(CH_3_-C), 10% δ(O=C-N), 20% other
IV	770–625	40% δ(O=C-N), 60% other
V	800–640	γ(N-H)
VI	600–540	γ(C=O)
VII	200	skeletal mode

^1^ ν—stretching, δ—bending, γ—out-of-plane bending

**Table 2 molecules-25-02498-t002:** IR marker bands of abnormal protein aggregation.

Marker Band of the Aggregation [cm^−1^]	Assignment	Peptide		Reference
amide I (1700-1600 cm^−1^)				
1695-1685/1633-1623	antiparallel β-sheet	amyloid β_1-42_	oligomers	[40,41]
		amyloid β_1-40_		[42,43,44]
		*α*-synuclein		[28,47,48]
		PrP_82–146_		[51]
1693-1685/1623–1613	antiparallel β-sheet	amyloid β_11-28_ fragment and its mutants in 21-23 position		[31]
1692/1620	antiparallel β-sheet	HET_218–289_		[55]
1691/1630	antiparallel β-sheet	Aβ 42CC oligomers/protofibrils		[32]
1688/1620	antiparallel β-sheet	human lysozyme, oligomers		[29]
1686/1616	antiparallel β-sheet	transthyretin (TTR) soluble aggregates		[52]
1686/1614	antiparallel β-sheet	hen egg white lysozyme (HEWL)		[56]
1684/1616		β2-microglobulin (short curved structures)		[54]
1684/1612	antiparallel β-sheet	SH3 domain, amorphous aggregates, non-fibrilar		[57]
1683/1612 ↑→	antiparallel β-sheet	insulin	oligomer	[39]
1678-1670	β-turns	amyloid β_11-28_ fragment and its mutants in 21-23 position		[31]
1670	β-turns	amyloid β_1-42_	oligomers and fibrils	[40]
		*α*-synuclein		[28]
1669	β-turns	HET_218–289_		[55]
1667-1661	3_10_-helix	E22K and A21G mutants of Aβ(11-28) fragment		[31]
1664	β-turns	SH3 fibrils/pepsin digested		[57]
1660-1650	random coil and/or helical structures	amyloid β_1-42_	oligomers and fibrils	[40]
		*α*-synuclein		[28]
1659-1652	α-helix	amyloid β_11-28_ fragment and its mutants in 21-23 position		[31]
1658	Turns	human lysozyme	monomers, oligomers, fibrils	[29]
1655	random coil	HET_218–289_		[55]
1649	unstructured	SH3 amorphous aggregates		[57]
1648-1639	random coil	amyloid β_11-28_ fragment and its mutants in 21-23 position		[31]
1648	random coil	PrP_82–146_		[51]
1644-1641	disordered/loops	human lysozyme	oligomers, fibrils	[29]
1641	disordered structures	SH3 fibrils/pepsin digested		[57]
1635-1624	β-sheet	amyloid β_11-28_ fragment and its mutants in 21-23 position		[31]
1633	parallel β-sheet	Sup35 crystals, prion-like		[58]
1630	parallel β-sheets	HET_218–289_		[55]
1630-1623	parallel β-sheet	amyloid β_1-42_	fibrils	[40,41]
		amyloid β_1-40_		[42,43,44]
1630-1614	parallel β-sheet	human lysozyme	fibrils	[29]
1628	parallel β-sheet	*α*-synuclein	fibrils	[28]
1626 ↑	parallel β-sheet	PrP_82–146_	fibrils	[51]
1626 ↑→	parallel β-sheet	insulin	fibrils	[39]
1625	parallel β-sheet	transthyretin (TTR)	fibrils	[52]
1620-1618		β2-microglobulin, fibrils		[54]
1620-1600	β-sheets	hen egg white lysozyme (HEWL)		[56]
1618 ←	parallel β-sheet	SH3 fibrils/pepsin digested		[57]

↑ increase in intensity, ← shift towards higher wavenumbers after aggregation, → shift towards lower wavenumbers after aggregation.

**Table 3 molecules-25-02498-t003:** AFM-IR marker bands of the neurodegenerative peptide aggregation.

Marker Band of the Aggregation [cm^−1^]	Assignment	Peptide	Reference
**amide I (1730–1600 cm^−1^)**			
1730	C=O	Boc-FF, FF	[72,73]
1700–1690	anti-parallel β-sheet	amyloid β (AβpE3-42)	[69]
1700–1600	ν(C=O) bk	unexpanded Exon1 (22Q)	[26,34,35,67,74]
1696–1690	anti-parallel β-sheet, carbamate group	Boc-FF, FF	[72,73]
1695–1665	β-turn and antiparallel β-sheets	ataxin-3	[38,66,70,71,75]
1695,1684	β-turn, anti-parallel β-sheet	amyloid β (AβpE3-42)	[69]
1695	β-turn	insulin	[68]
1692	anti-parallel β-sheet	amyloid β with 5% pyroglutamylated peptide	[69]
1692	anti-parallel β-sheet	unexpanded Exon1 (22Q)	[26,34,35,67,74]
1689,1625	anti-parallel β-sheet	mutant oligomer G37C	[65,76]
1684	glutamine side chain vibrations and β-turn	expanded Exon1 (42Q)	[26,34,35,67,74]
1664,1655	α-helix, 3-helix	Boc-FF, FF	[72,73]
1662	β-turn	amyloid β	[65,76]
1660–1650	α- helix	ataxin-3	[38,66,70,71,75]
1660	α-helix/unordered protein secondary structures	Insulin	[68]
1658	poor α-helix	amyloid β (AβpE3-42)	[69]
1658	α-helix	expanded Exon1 (42Q)	[26,34,35,67,74]
1645–1630	random coil	ataxin-3	[38,66,70,71,75]
1645–1635	random coil	unexpanded Exon1 (22Q)	[26,34,35,67,74]
1640–1600	residual water absorption, NH_3_^+^ group	Boc-FF, FF	[72,73]
1638	random coil	amyloid β (AβpE3-42), amyloid β with 5% pyroglutamylated peptide	[69]
1635–1610	low density native/high density amyloid β-sheets	ataxin-3	[38,66,70,71,75]
1635	β-sheet	unexpanded Exon1 (22Q), expanded Exon1 (42Q)	[26,34,35,67,74]
1631	parallel β-sheet secondary structure	amyloid β	[65,76]
1623	high β-sheet	amyloid β (AβpE3-42)	[69]
1620	β-sheet	Insulin	[68]
**amide II (1600–1500 cm^−1^)**			
1580–1510	bk δ(N-H), ν(C-N)	ataxin-3	[38,66,70,71,75]
1555,1520	NH vibrations	Boc-FF, FF	[72,73]
**C-C ring vibrations**			
1605,1495,1452,1430	C-C ring vibrations	FF	[72,73]
**amide III (1400–1200 cm^−1^)**			
1350–1200	ν(C-N), δ(N-H),ν(C-C), δ(C=O)	ataxin-3	[38,66,70,71,75]

ν—stretching, δ—bending, bk—backbone.

**Table 4 molecules-25-02498-t004:** Raman marker bands of peptide aggregation.

Marker Band of the Aggregation ^1^	Assignment	Peptide	Reference
**amide I (1700–1600 cm^−1^)**			
1690–1600 ↑1672→	β-sheets formation	insulin, hen egg white lysozyme (HEWL)	[85,86,88]
**amide II (1580–1480 cm^−1^)**			
1580–1480 ↑	β-sheets formation	HEWL	[84,85,86]
1550→		1-SS-carboxymethyl lactalbulin (1-SS-LA), HEWL	
**Cα H**			
1390 ↑		insulin, HEWL	[85,86,88]
**amide III (1283–1218 cm^−1^)**			
1320–1270 ↑1270–1230 ↓		HEWL	[88]
**disulfide (S-S) (550–450 cm^−1^)**			
523, 507 ↓490		HEWL	[84,85]
540, 510 ↓508		apo-α-lactabulin (LA)	
**phenylalanine (Phe)**			
1000 ↓		HEWL	[87,88]

*^1^* ↑ an increase of peak intensity, ↓ a decrease of peak intensity, → a peak shift to higher energy

**Table 5 molecules-25-02498-t005:** SERS marker bands of the neurodegenerative peptide aggregation.

Marker Band of the Aggregation	Assignment	Peptide	Reference
**amide I (1700–1600 cm^−1^)**			
1678–1664↑	β-sheets	insulin	[99]
1664–1640↓	α-helics, unordered		
**amide III (1283−1218 cm^−1^)**			
1244 ↑	β-sheets	amyloid β_1–42_	[95,96]
1266 ↓	α-helics		
**CCN stretching**			
1144 ↓		amyloid β_1–42_	[96]
**C-C stretching**			
960 ↑		amyloid β_1–42_	[95]

↓ a decrease of peak intensity, ↑an increase of peak intensity

**Table 6 molecules-25-02498-t006:** TER marker bands of the neurodegenerative peptide aggregation.

Marker Band of the Aggregation [cm^−1^]	Assignment	Peptide	Reference
**amide I (1700–1600 cm^−1^)**			
1680–1660 ↑	β-sheets	amyloid β_1-42_, insulin,	[111,113,115,116,117,118]
		β2-microglobulin	
1674 ↑	β-sheets	hIAPP (Amylin)	[112]
1655–1630↓	random coil	amyloid β_1-42_	[117]
1640↓,1664–1640 ↓	α-helics, unordered	hIAPP (amylin), insulin	[112,113,115,118]
**Cα H/N-H (1370–1360 cm^−1^)**			
1364 ↑	β-sheets	amyloid β_1-42_	[116]
**amide III (1283–1218 cm^−1^)**			
1228–1218 ↑	parallel β-sheets	amyloid β_1-42_	[117]
1242–1233, 1250↑	antiparallel β-sheets	insulin	[116,117]
1235–1230↑	β-sheets	amyloid β_1-42,_	[111,115]
1261–1248, 1258 ↓	random coil	β2-microglobulin	[116,117]

↓ a decrease of peak intensity with the ongoing fibrillation, ↑ an increase of peak intensity with the ongoing fibrillation.

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
