# Peer review of "Molecular Spectroscopic Markers of Abnormal Protein Aggregation"

_molecules, 2020, doi:10.3390/molecules25112498_

Round 1
Reviewer 1 Report
The manuscript of Wilkosz et al. reviewed the applications of vibrational spectroscopies, including FTIR and Raman in studying protein misfolding and aggregation. These molecular spectroscopies are particularly sensitive to the secondary structural changes along the protein aggregation pathway. The fundamentals of these methods were briefly summarized and the recent progress of using these techniques is introduced. More specifically, the application of more sensitive molecular spectroscopic technologies, such as Nano-FTIR, AFM-IR, SERS, and TERS was included. The manuscript is well organized and well written, and suitable for publication after considering below some minor concerns.
- FTIR is an old and commonly used technique in studying protein secondary structures. Recent advances of FTIR, including 2D-IR and local vibrational labels in studying protein amyloid, may worth being briefly introduced.
- Line 71-72: Abeta native molecule has been proved to be mainly disordered.
- Table 4: bottom line, the peak of 1000 should be decreasing upon aggregation.
- Gramma mistake needs to be fixed. For example: Line 475-"understand" should be "understood"; Line 477-"from the other had"
Author Response
Dear Reviewer,
We would like to thank you for all the remarks and suggestions, which helped us to improve the quality of the manuscript. We have introduced all of the suggested corrections into the manuscript as briefly summarised below.
For your convenience we have uploaded a marked version of the manuscript highlighting all changes.
1) The manuscript has been improved, please see page 5 and page 6. References were also revised. Following sentences were added:
"Another recent work concerning α-synuclein aggregation describes the influence of the ionic strength on the β-sheets orientation in fibrils studied with 1D- and 2D-IR spectroscopy [49]. 2D-IR spectroscopy provides information about the specific residues of interest and is sensitive to more ordered structures in general [49,50]. The β-sheet arrangement in fibrils turned out to be correlated with salt concentration during fibrilization. α-synuclein aggregation in low ionic-strength conditions (NaCl concentration ≤ 25 mM) results in parallel β-sheet orientation in fibrils, while the fibrilization upon high salt concentration, including physiological conditions, contributes to the antiparallel β-sheet arrangement."
"The aggregation kinetics of human islet amyloid polypeptide (hIAPP) was studied using 2D infrared spectroscopy combined with site-specific isotope labeling [59]. This methodology allowed to follow the intensity growth of the 1617 cm-1 peak related to the increase of β-sheet content upon aggregation and fibrils formation. Ami et al. [60] incorporated the FTIR microscopy to study aggregates of amyloidogenic immunoglobulin light chains (LCs) occurring in the light chain (AL) amyloidosis pathology. The applied methodology involved the FTIR in situ studies of unfixed tissues (hear and subcutaneous abdominal fat) derived from AL amyloidosis affected patients as well as the research of in vitro aggregated peptide (derived from a patient). The infrared β-sheet signature characteristic for amyloid aggregation was possible to detect in situ in the spectra of tissues."
2) Corrected, please see page 2
" Native Aβ protein has mainly disordered structure but a number of research reveals the presence of local regions displaying a secondary structure [16,17]."
3) Corrected, please see table 4.
4) Corrected.
5) As suggested English was revised.
Reviewer 2 Report
In the accompanying manuscript, the authors give a comprehensive overview of the use of Raman and infrared spectroscopy for the study of protein aggregation. Their work focuses on the nanoscale spatial resolution of tip-enhanced Raman and infrared nanospectroscopy and the high sensitivity of surface-enhanced Raman spectroscopy, which brought new insights into our knowledge of abnormal protein aggregation.
They combined all nano- and microspectroscopic marker bands associated with abnormal aggregation. Each part introduces the above mentioned physical principles of each spectroscopic technique and briefly and concisely describes all spectral markers detected by these techniques in the spectra of neurodegenerative proteins and their model systems. Finally, a section on the application of multivariate data analysis for the extraction of spectral marker bands is included.
It's a well-written review.
Author Response
Dear Reviewer,
we would like to thank you for your opinion. The suggested moderate English changes were performed.
For your convenience we have uploaded a marked version of the manuscript highlighting all the modifications.
Sincerely,
Ewelina Lipiec and Kamila Sofińska